# Proprioceptive and Strength Exercise Guidelines to Prevent Falls in the Elderly Related to Biomechanical Movement Characteristics

**DOI:** 10.3390/healthcare12020186

**Published:** 2024-01-12

**Authors:** Pedro Aleixo, João Abrantes

**Affiliations:** 1Centro de Investigação em Desporto, Educação Física, Exercício e Saúde (CIDEFES), Universidade Lusófona, 1749-024 Lisbon, Portugal; 2Centre for Research in Applied Communication, Culture, and New Technologies (CICANT), Universidade Lusófona, 1749-024 Lisbon, Portugal; joao.mcs.abrantes@ulusofona.pt

**Keywords:** elderly, falls, stability, articular mobility, proprioceptive exercise, strength exercise, fall prevention program

## Abstract

Falls are a major concern in the elderly and walking is an important daily activity in which falls occur, with tripping and slipping being the most frequent causes. Gait biomechanical parameters have been related to the occurrence of falls in the elderly. Moreover, there is evidence that falls can be prevented through exercise programs, which have been shown to be also effective in improving gait biomechanical parameters. However, a question remains: “What types of exercises must be included in exercise programs to prevent falls?”. The purpose of this manuscript was to present guidelines for a fall prevention exercise program for the elderly, which was created with the aim of improving the gait biomechanical parameters related to falls. The critical review performed during the preparation of this manuscript collected important evidence and knowledge in order to create a structural basis for the development of a fall prevention exercise program. This type of program should last 6 or more weeks and be prescribed based on four movement pillars (locomotion, level changes, pulling and pushing, and rotations); however, the locomotion pillar must be the focus of the program. Proprioceptive and functional strength exercises should be included in this program. Based on the theoretical rationale, a proposal for a fall prevention exercise program is presented.

## 1. Introduction

The World Health Organization classifies the elderly as subjects aged 60 or over [1], although there are other classifications such as the ACSM definition that considers an elderly adult to be more than 65 years old [2]. The elderly show the highest incidence and the worst consequences of falls [3]. The consequence of falls in this population involves a lower quality of life, a greater fear of falling, loss of confidence, and fall-related fractures [4,5,6]. Indeed, falls are the second leading cause of unintentional injury deaths worldwide [7]. Additionally, the associated economic burden of this important public health problem for families, communities, and society cannot be ignored [8].

Falls are dependent on the subject’s condition and extrinsic factors. The factors related to the subject’s condition include muscle weakness, functional and stability disorders, cognitive impairment, and visual deficits; factors extrinsic to the subject or associated with the involvement include medication, poor lighting, and rugs that slide [9]. Consequently, several interventions have been developed in order to reduce these risks, e.g., exercise programs, educational programs, medication optimization, and environmental modification [10]. Some of those factors, like muscle weakness or functional and stability disorders, can be modified using exercise programs, whereas others, like medication or visual deficits, require different interventions [11]. Certainly, interventions comprising exercise programs can prevent falls in the elderly [12].

An important first approach to prevent falls is related to gait. In fact, it is well known that walking is an important daily activity in which falls occur, with tripping and slipping being the most frequent causes [13]. Increased knowledge of fall-related gait biomechanics can help in the development of interventions to reduce their occurrence [14]. In this way, methods of study based on laboratory tools allow more objective assessments [15]. Actually, assessments conducted in the laboratory have associated some gait biomechanical parameters with the occurrence of falls in the elderly [16,17,18,19,20,21,22,23,24,25,26,27]. Thus, the gait biomechanical parameters related to falls must be improved in order to prevent falls. Additionally, there is evidence that falls can be prevented through exercise programs [12], which have been shown to be effective in improving the aforementioned gait biomechanical parameters [28,29,30,31,32]. The evidence collected in the referenced studies was a structural basis for the development of an exercise program to prevent falls. In this way, the purpose of this manuscript was to present guidelines for a fall prevention exercise program for the elderly, which was created with the aim of improving the gait biomechanical parameters related to falls. Therefore, an explanation of its theoretical rationale and development was included.

## 2. Theoretical Rationale of the Exercise Program

A recent systematic review [12], which included 108 randomized controlled trials, pointed to high-certainty evidence concerning the efficacy of exercise programs in reducing the rate of falls and the number of elderly who experience them. However, a question remains: “What types of exercises must be included in exercise programs to prevent falls?”. According to the same systematic review, different types of exercise yielded diverse impacts on the occurrence of falls in the elderly. Thus, the exercise programs most likely to reduce falls include multiple types of exercise, i.e., exercises targeting stability, functional exercises, and strength exercises. Furthermore, the authors also found that challenging locomotion exercises, such as Tai Chi, can also prevent falls. Therefore, an exercise program to prevent falls must include all these types of exercise.

### 2.1. Exercise Prescription Based on Improving Movement Quality

An important point of this rationale, perhaps crucial in its understanding, is to realize the extreme importance of efficient movement control for fall avoidance. On the one hand, impaired movement control can lead to falls during daily activities [33]. On the other hand, inefficient movement is metabolically expensive, meaning the onset of fatigue will be faster and the subject will be less able to perform daily living tasks [34]. Thus, improving movement means not only an improvement in performance and achievement of the task objective, i.e., efficacy, but also regarding its economy, i.e., efficiency. Effective and efficient movement, i.e., the definition of movement quality, depends on two fundamental capacities, namely stability and joint mobility [35,36]. Joint mobility is the ability to reach the required joint position for effective and efficient movement performance [37], while stability can be divided into postural and joint stability. Postural stability is the ability of the entire body to preserve its state of tendency towards equilibrium using its own means of motor control, whereas joint stability results from the motor ability to control the elements that act on each joint complex in order to maintain the proper angular position, providing reliability of execution [38]. Evidently, these two stabilities are interdependent since both are controlled by the central nervous system. In fact, the central nervous system controls joint mobility as well as postural and joint stabilities through three distinct but interdependent levels: higher level—cerebral cortex; middle level—brain stem; lower level—spinal cord [39]. The cerebral cortex controls the more complex voluntary movements, and the brain stem contains the great circuits that control postural stability and many of the stereotyped body movements [39]. The reciprocal innervation mechanism is an important process of the spinal cord movement control, coordinating the actions of agonist and antagonist muscles [40]. This mechanism is dependent on the quality of proprioception, a subcomponent of the somatosensory information that includes the afferent information arriving from the mechanoreceptors located at the periphery, i.e., Golgi tendon organ, neuromuscular spindle, and joint receptors [39]. This information is integrated into the spinal cord in order to control joint mobility and joint stability [40]. Still, this information also arrives at the higher levels of the central nervous system, namely the brain stem, which is quite essential regarding postural stability control [41]. Postural stability control, at the brain stem, is also dependent on vision and vestibular information; however, the information from peripheral mechanoreceptors is the most important from a clinical orthopaedic perspective [39]. In this way, proprioception has an important role in joint mobility as well as in postural and joint stabilities [28,41]. Proprioceptive exercises aim to improve the efficacy of the afferent feedback, in order to attain functional segment control and to achieve appropriate neuromuscular control of the muscles encompassing joint complexes [42]. They distinguish themselves from others by a greater ability to stimulate and enhance proprioception and somatosensory function. Thus, proprioceptive training programs may be an efficient tool to improve the agonist/antagonist muscle communication [42]. According to a systematic review that investigated the effectiveness of proprioceptive exercise programs for improving motor function [43], programs lasting 6 or more weeks can lead to improvements in proprioception and somatosensory function. Nonetheless, the optimal dose–response of this type of exercise is yet to be defined due to the enormous variability and lack of detail observed in the various studies concerning the training parameters, e.g., weekly frequency and workout duration. Despite this fact, it is widely accepted that stimuli applied to nervous structures during exercise will be more effective in conditions of absence of fatigue [44]. Moreover, the American College of Sports Medicine (ACSM) postulated that proprioceptive training is effective in reducing and preventing falls if performed 2–3 times/week [2].

Since exercise programs should aim to improve movement, namely through improving joint mobility and stability, it is essential to realize this is associated with the development of processes that underlie motor control, with the neuromuscular system being the key structure for all these processes. In this way, it is necessary to understand that muscles are just the “workers” and movement quality depends mainly on neuromuscular coordination, which takes place at two different levels, i.e., intramuscular coordination and intermuscular coordination. On the one hand, the central nervous system modulates the duration and intensity of activation of each muscle involved in the movement through intramuscular coordination, and on the other, ensures the combined and complementary intervention of the various muscles involved in the movement through intermuscular coordination [45]; muscular work is completely dependent on this control by the central nervous system. Given the above, it is not surprising that exercise prescription should be founded on improving movement patterns. Wickstrom was one of the first to define the basic movement patterns: walking, running, jumping, throwing, catching, striking, and kicking [46]. The author’s perspective was quite focused on the field of motor development during childhood. Since then, other authors have also presented their perspectives on this issue; however, more recently, two views have stood out, namely those of Santana and the Gray Institute [47,48], both focused on exercise prescription. Thus, the Gray Institute defines the following movement patterns: lunging—action of taking a step in a certain direction and returning to the starting position; squatting—action of squatting and returning to the starting position; jumping—action of jumping; reaching—action of reaching an object; lifting—action of changing the vertical position of an object; pushing—action of pushing an object; pulling—action of pulling an object; gait—action corresponding to locomotion, walking, or running. On the other hand, Santana classifies human movements into four movement pillars: locomotion, level changes in the subject’s centre of mass, pulling and pushing with upper limbs, and rotations. Locomotion is one of the most basic movement skills and is essential during daily living activities. Level changes are characterized by movements of the trunk or lower limbs, or a combination of both, that vertically displace the centre of mass. Pulling is any movement that brings the elbows or hands inward or toward the main line of the body. Pushing is any movement that brings the elbows or hands outward or away from the main line of the body. Finally, rotation is the most important pillar concerning daily living activities, involving movements that occur in the transverse plane. Both approaches postulated that exercise programs should be prescribed based on these movement patterns. Although any classification of movement patterns is valid and applicable to training, Santana’s proposal is easier to implement from the point of view of training organization and planning.

Santana also presented the concept of functional strength, which was defined as the amount of strength a subject can use during daily living activities and, according to the author, the most important strength to develop regarding functionality in everyday tasks [47]. According to other authors [49,50], functional strength training should focus on the quality of the movement pattern and treat functional movement as a priority. Therefore, exercises that aim to increase functional strength should be prescribed with low loads and with a focus on monitoring movement quality. Several research works studied the effects of this type of training and found improvements regarding joint mobility, postural stability, strength, and power in untrained young girls [50]; joint mobility, postural stability, and power in young tennis players [51]; agility, strength, and power in young adults [52]; postural stability and coordination in trained young males [53]; postural stability in young adult soccer players [54]; postural stability and strength in young adults [55]; joint mobility and postural stability in middle-aged and elderly adults [49]; strength, gait speed, and functional capacity in disabled elderly [56]; gait spatial and temporal parameters in elderly with dementia [57]. On the other hand, some of these studies found superior benefits of a functional strength training program when compared with a traditional strength training program concerning joint mobility, postural stability, strength, and power [50]; joint mobility, postural stability, and power [51]; agility, strength, and power [52]; postural stability and coordination [53]; gait speed and functional capacity [56]. Finally, according to Santana, it is important to develop functional strength through exercises for the four pillars of movement.

### 2.2. Workout Session for the Elderly—Guidelines

The National Academy of Sports Medicine (NASM) postulated that exercise progression with the elderly should be slow, well monitored, and based on postural control [58]. Any workout session, especially with the elderly, must consider the following structure: warm-up; the main part; cool-down [2,58]. During warm-up, it is vital to prepare the elderly for the upcoming movements of the main part of the session. In this way, active stretching, i.e., range of motion achieved using the strength of the antagonist muscles of those that are stretched [2], is an essential warm-up exercise to activate the nervous system [59]. During the main part of the session, exercises should be prescribed in order to achieve the stimuli defined for the workout session [2,58]; in our case, improving the gait biomechanical parameters related to falls and the four movement pillars, i.e., locomotion, level changes in subject’s centre of mass, pulling and pushing with upper limbs, and rotations. According to the most recent ACSM guidelines [2], strength training in the elderly should have a frequency of ≥2 workouts/week, 8–10 exercises/session, ≥1 set/exercise of 10–15 repetitions, an intensity of 40–50% of the one repetition maximum or at a 5–6 level of a perceived 10-point physical exertion scale (for seniors who are starting a strength program). During cool-down, it is important to promote recovery and return the body to a pre-workout level [2,58]. NASM and ACSM suggested static stretching for this purpose [2,58].

### 2.3. Gait Biomechanical Parameters Related to Falls

As mentioned in the introduction, some gait biomechanical parameters have been related to the occurrence of falls in the elderly. A well-studied biomechanical parameter regarding tripping is minimum foot clearance (MFC), i.e., the minimum vertical distance between the lowest point of the foot of the swing leg and the walking surface during the swing phase of the gait cycle [16]; however, several studies used the toe as the reference point, defining the term minimum toe clearance. According to a systematic review [16], there seemed to be no differences in MFC values when comparing elderly fallers with non-fallers. Nevertheless, data from the same review showed that elderly fallers yielded a higher MFC variability—standard deviation used to assess variability. A previous study pointed out that the higher variability yielded by the elderly reflected impaired motor control and the consequent increased risk of trip-related falls (standard deviation was used to assess variability) [60].

Lower-limb kinematics is strictly related to two important events during gait, i.e., MFC and the heel strike [61]. As described in the previous paragraph, MFC is an event associated with trip-related falls, while the heel strike, i.e., the instant when the heel or foot makes initial contact with the ground [62], is an event related to slip-related falls [17]. In this way, higher heel horizontal velocity at heel strike may increase the potential for a slip-induced fall [17,62]. Moreover, the angle between the foot and the floor at the heel strike is another parameter associated with slip-related falls [17,20]. Therefore, an upper angle between the foot and the floor at the heel strike increases the risk of fall due to a reduction in the shoe–floor contact area and an increased braking impulse at landing [17].

Foot control is also quite vital regarding articular and postural stabilities during the gait stance phase [63]. This foot control can be divided into three important sub-phases with different aims each: (1) controlled plantar flexion sub-phase—to control the foot impact with the ground; it is strictly related to the heel strike event; starts at the heel strike and ends at the instant of occurrence of maximum plantar flexion; (2) controlled dorsiflexion sub-phase—to control the foot so that it remains stable and allows the body to move forward; starts at the end of the previous sub-phase and ends at the instant of occurrence of maximum dorsiflexion; (3) powered plantar flexion sub-phase—to control the foot in order to properly push the lower limb into a stable swing phase; starts at the end of the previous sub-phase and ends at toe-off [63,64].

Ankle stability presents itself as the crucial capacity during the controlled dorsiflexion sub-phase. Joint stiffness plays a key role in achieving adjusted joint stability during movement [65]. In order to determine joint stiffness, the literature presents dynamic joint stiffness (Nm/Kg/°) as the reference parameter [36]. Based on this parameter, non-faller post-menopausal women with rheumatoid arthritis yielded stiffer behaviour compared to fallers during the controlled dorsiflexion sub-phase [21].

The powered plantar flexion sub-phase is also an important period regarding the gait cycle [63,64]. The ankle power peak during this sub-phase can be used to differentiate fallers from non-fallers in post-menopausal women with osteoporosis [22] and with rheumatoid arthritis [23]: both faller populations yielded a lower ankle power peak. Furthermore, the ankle moment of force peak during the same sub-phase is also reduced in rheumatoid arthritis post-menopausal women fallers compared to non-fallers [23].

Elderly fallers, compared to non-fallers, also yielded different values of the gait spatial and temporal parameters, i.e., shorter step/stride length [24,25] and lower cadence [25], leading to a lower gait speed [24,25,26,27]; as well as a shorter single-support phase [25] and a longer double-support phase [24,25,27]. Furthermore, a higher variability in the spatial and temporal parameters has also been found in elderly fallers, namely in the following parameters: stride length [27]; step length and double-support phase [24]; swing time and stride time [18,27]. In the elderly, a lower gait speed, a shorter stride length, and a longer double-support time were clearly related to fear of falling but presented slight evidence of an independent association with prospective falls [60]. On the other hand, increased variability in the gait spatial–temporal parameters (i.e., gait speed, stride length, and double-support time) was related to prospective falls in the elderly but presented no relation to fear of falling. Thus, the variability in gait spatial and temporal parameters measured through the standard deviation can be an important parameter for recognising subjects at high risk for falls and for assessing preventive interventions.

In numerous daily situations, the elderly may need to take recovery steps in order to maintain their postural stability, e.g., when turning suddenly to the side while standing or being jostled in a crowd [66]. Therefore, a recovery step can be used to restore postural stability when that is lost during daily living activities demanding a standing position. From a biomechanical point of view, this strategy tries to maintain or recover the position of the centre of mass over the base of support. According to recent research [19], recurrent fallers yielded impaired kinematics during lateral step recovery responses compared to non-fallers, i.e., higher centre of mass displacement, longer step initiation duration, and longer step duration.

The increase in elderly functionality should encompass improvements in the gait biomechanical parameters mentioned in the previous paragraphs. In this way, strength exercise programs [31,32] as well as proprioceptive exercise programs [28,29,30] showed an ability to improve these parameters. According to a previous study [57], improvements in gait spatial and temporal parameters of the elderly with dementia were found as a result of a functional strength training program. The authors concluded that the exercise program may represent a model for preventing and rehabilitating gait deficits in the target group.

In summary, the three sub-phases of the gait stance phase are important periods regarding the occurrence of falls during gait, as well as the heel strike and MFC events. Diverse gait biomechanical parameters during these periods and events were related to falls, which can and must be improved through exercise programs. Therefore, the following gait biomechanical parameters were mentioned as associated with the occurrence of falls among the elderly: (1) higher MFC variability; (2) higher heel horizontal velocity at heel strike; (3) higher angle between foot and floor at heel strike; (4) lower dynamic joint stiffness during the controlled dorsiflexion sub-phase; (5) lower ankle power peak and ankle moment of force peak during the powered plantar flexion sub-phase of gait; (6) weakened spatial and temporal parameters, i.e., shorter step/stride length, lower cadence, lower gait speed, shorter single-support phase, and longer double-support phase; (7) higher variability in the spatial and temporal parameters, i.e., gait speed, stride length, and double-support time; (8) impaired kinematics during lateral step recovery. Several of these parameters are dependent on motor control of the lower limb joints, i.e., dependent on joint mobility and joint stability. An example of that is the dependence of MFC on the angles of the swing lower limb joints, i.e., hip, knee, and ankle [67]. Otherwise, the angle between the foot and the floor at the heel strike as well as the ankle power peak and moment of force peak during the powered plantar flexion sub-phase are also dependent on the movement control of the ankle. Moreover, one way to reduce the horizontal velocity of the foot at heel strike is through increased and/or an earlier activation of the hamstrings [62]. Once again, this parameter is dependent on motor control of the lower limbs. According to the abovementioned, an exercise program aiming to prevent falls must consider exercises that stimulate the motor control of lower limb joints. Additionally, exercise programs should also promote confidence and reduction of fear of falling, “working out” the improvement of the spatial and temporal parameters. Finally, exercises comprising lateral perturbation may enhance the step recovery and consequently the quality of fall prevention programs [66].

### 2.4. Theoretical Rationale Summary

In summary, an exercise program to prevent falls in the elderly should last 6 or more weeks. It must be prescribed based on the four movement pillars (i.e., locomotion, level changes, pulling and pushing, and rotations); however, the focus of the program must be the locomotion pillar and on improving the biomechanical parameters related to falls and associated with the important gait events. Thus, proprioceptive and functional strength exercises should be included in this program.

## 3. Exercise Program to Prevent Falls—Proposal

In the next paragraphs, an exercise program to prevent falls in the elderly is proposed, which is based on the theoretical rationale explained in the previous section of this manuscript. This program is not intended to be a recipe, but rather a practical example of its feasibility in the field.

An exercise program to prevent falls in the elderly should last at least 12 weeks, with three workouts/week (weekly frequency)—not performed on consecutive days. However, it must be maintained over time to avoid the loss of acquired capabilities. The session should last about 45 min. The following session structure is proposed: 5 min with active stretching exercises (warm-up); 20 min with proprioceptive exercises focused on the locomotion pillar, which is effective in improving gait biomechanical parameters associated with falls [28,29]; 15 min with functional strength exercises that aim to improve the other three pillars; 5 min with static stretching exercises (cool-down).

The active stretching exercises should prepare the elderly for the main part of the workout session. Thus, during the first 5 min ± 12 exercises should be performed, stretching all areas of the body—15 s/stretch. The elderly should be focused on performing active stretching. Examples of these exercises can be viewed at http://elderly.falls.ulusofona.pt/ (accessed on 28 November 2023) [68].

During the 20 min allocated to the proprioceptive exercises, it is proposed the elderly perform nine exercises to improve the gait biomechanical parameters mentioned in a previous point. These exercises can be viewed at http://elderly.falls.ulusofona.pt/ [68].

The first four exercises should be included in all workout sessions: (1) in single leg stand position, perform plantar flexion and dorsiflexion of the swing ankle—three sets for each side with six repetitions each set; alternating sides between sets; (2) in single leg stand position, perform extension and flexion of the swing knee—three sets for each side with six repetitions each set; alternating sides between sets; (3) in single leg stand position, perform flexion and extension of the swing hip—three sets for each side with six repetitions each set; alternating sides between sets; (4) step forward and backwards—three sets for each side with three repetitions each set; alternating sides between sets [68]. These four exercises aim to improve proprioception related to postural stability and local motor control of the lower limb joints. The other five exercises should be selected according to their level of complexity and the subject’s capacity to perform them—three sets for each side with three repetitions each set (alternating sides between sets).

The fifth exercise aims to improve MFC. In this way, one of the following exercises should be selected (which increases their level of complexity): (1) step forward and backwards with an exaggerated hip flexion; (2) step forward and backwards over an obstacle, e.g., a step; (3) step forward and backwards over an obstacle, e.g., a step, with an exaggerated hip flexion; (4) lunge followed by a step forward with exaggerated hip flexion, and then a step backwards to the lunge initial position [68].

The sixth exercise aims to improve ankle control. In this way, one of the following exercises should be selected (which increases their level of complexity): (1) step forward and backwards with plantar flexion and dorsiflexion during step; (2) step forward and backwards over an obstacle, e.g., a step, with plantar flexion and dorsiflexion during step; (3) lunge followed by a step forward with plantar flexion and dorsiflexion during the step, and then a step backwards to the lunge initial position [68].

The seventh exercise aims to improve ankle control during heel strikes. In this way, one of the following exercises should be selected (which increase their level of complexity): (1) step forward and backwards controlling the heel strike—reducing the speed of the swing foot; (2) step forward and backwards over an obstacle, e.g., a step, controlling the heel strike—reducing the speed of the swing foot; (3) in single leg stand position with the other foot in position to make the heel strike; knee flexion of the stance leg until the instant before heel strike—focus on the angle between the swing foot and the floor; (4) lunge followed by a step forward controlling the heel strike—reducing the speed of the swing foot—and then a step backwards to the lunge initial position [68].

The eighth exercise aims to improve ankle power during the powered plantar flexion sub-phase of gait. In this way, one of the following exercises should be selected (which increases their level of complexity): (1) lunge—one leg is positioned forward and the other is positioned behind; knee flexion of the forward leg; then, knee extension of the forward leg until the initial position; (2) lunge followed by a step forward, and then a step backwards to the lunge initial position; (3) sequential lunges followed by a step forward; (4) sequential lunges followed by a step forward, with exaggerated hip flexion, controlling the heel strike, and with plantar flexion and dorsiflexion during step [68].

The ninth exercise aims to improve the lateral step recovery. In this way, one of the following exercises should be selected (which increases their level of complexity): (1) lateral step controlling the foot strike; (2) lateral step with lunging; (3) sequential lateral lunges followed by a lateral step [68].

Another way to regulate the complexity of these exercises is by increasing, or decreasing, step amplitude. This will help to improve confidence and the spatial and temporal parameters. During the 12-week exercise program, the elderly will be challenged to increase exercise complexity whenever they perform them easily.

During the 15 min assigned to the functional strength exercises, the elderly should perform four exercises using body weight or inexpensive equipment, i.e., medicine balls, rubber bands, and Swiss balls. These exercises can be viewed at http://elderly.falls.ulusofona.pt/ [68]. They should be performed in a circuit, twice, according to the following sequence: (1) exercise for the level changes pillar; (2) exercise for pushing the pillar; (3) exercise for the pulling pillar; (4) exercise for the rotations pillar. They should be selected according to their complexity level and the ability that each participant will have in carrying them out—10 repetitions/exercise.

The first exercise should be selected from one of the following exercises (which increase their level of complexity): (1) squat with back supported on a Swiss ball; (2) squat; (3) squat with a medicinal ball in hands—moving the ball away from the body increases the exercise complexity [68].

The second exercise should be selected from one of the following exercises (which increase their level of complexity): (1) while sitting, pushing rubber bands; (2) while standing, pushing rubber bands; (3) while standing on foam, pushing rubber bands [68].

The third exercise should be selected from one of the following exercises (which increase their level of complexity): (1) while sitting, pulling rubber bands; (2) while standing, pulling rubber bands; (3) while standing on foam, pulling rubber bands [68].

The fourth exercise should be selected from one of the following exercises (which increase their level of complexity): (1) while standing, perform trunk rotations holding a medicinal ball—moving the ball away from the body increases the exercise complexity; (2) while standing, perform trunk rotations against rubber bands; (3) lying on a Swiss ball, perform trunk rotations against rubber bands [68].

The elderly should be challenged to increase exercise complexity whenever they perform them easily. Moreover, functional strength exercises should also be regulated using a perceived 10-point subjective exertion scale—between 5 and 6.

During the last 5 min, 12 static stretching exercises will be performed—15 s/stretching. All exercises can be viewed at http://elderly.falls.ulusofona.pt/ [68].

### Limitations of the Exercise Program Proposal and Special Considerations

The implementation of the exercise program proposal may encompass some difficulties for very frail elderly people. Nonetheless, remembering that this proposal is just that and not a recipe, in very frail elderly the exercise prescription should be based on four movement pillars, especially the locomotion pillar, even if this means reducing exercise complexity to a minimum—i.e., single-joint exercises with maximum stability. In this way, the optimal situation would be one-on-one sessions, e.g., sessions run by “personal trainers”. On the other hand, with three shorter sessions a week, it is also possible to organize groups with fewer people and according to specific needs (for psychomotor development, pathologies, etc.). The model must therefore be adapted to each of these groups, with less dispersion of attention on the part of the professional leading the session.

## 4. Conclusions and Future Investigation

A fall prevention exercise program for the elderly should last 6 or more weeks and be prescribed based on four movement pillars (locomotion, level changes, pulling and pushing, and rotations); however, the locomotion pillar must be the focus of the program, and proprioceptive and functional strength exercises should be included.

Future investigation should consider testing the guidelines presented regarding the proposed fall prevention exercise program for the elderly. In this way, the referred biomechanical gait parameters associated with falls should be evaluated using gold standard measures, i.e., three-dimensional motion analysis systems [69]. On the other hand, the duration of the program and exercises could be the subject of a study to justify the choices made and quantify the gains of the program.

## Data Availability

Not applicable.

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
