# Peer review of "Proprioceptive and Strength Exercise Guidelines to Prevent Falls in the Elderly Related to Biomechanical Movement Characteristics"

_healthcare, 2024, doi:10.3390/healthcare12020186_

Round 1

Reviewer 1 Report

Comments and Suggestions for Authors

The study presents a relevant contribution to the area and aims to present guidelines for a fall prevention exercise program in elderly, which was created with the aim of improving the gait biomechanical parameters related to falls. The manuscript is well written and structured and requires few revisions as follows:

Page 4. line 158. The authors describe the concept of functional strength and describe numerous benefits. However, several of these benefits are also well described in the literature for conventional strength training. What would be the existing gap? Insert an explanation for this counterpoint.

Page 7. Line 312. Why three workouts per week if the literature already shows that twice a week are enough to promote gains related to strength, functional capacity, balance and other parameters?

Present a limitations session, indicating whether this training would be suitable for all elderly people, considering different age groups, pathologies, etc.

Author Response

Dear reviewer.

Thank you very much for revisions and suggestions. We truly believe that this revisions and suggestions will improve the manuscript.

The study presents a relevant contribution to the area and aims to present guidelines for a fall prevention exercise program in elderly, which was created with the aim of improving the gait biomechanical parameters related to falls. The manuscript is well written and structured and requires few revisions as follows:

RESPONSE: Thank you for your comment.

Page 4. line 158. The authors describe the concept of functional strength and describe numerous benefits. However, several of these benefits are also well described in the literature for conventional strength training. What would be the existing gap? Insert an explanation for this counterpoint.

RESPONSE: Thank you for your comment. The authors agree with you that the benefits of the traditional strength training are well documented. However, the point is to have greater benefits with other types of training or methodologies. According to what is mentioned in the article, there seem to be greater benefits of functional strength training compared to traditional strength training: “On the other hand, some of these studies found superior benefits of a functional strength training program when compared with a traditional strength training program concerning: joint mobility, postural stability, strength, and power [50]; joint mobility, postural stability, and power [51]; agility, strength, and power [52]; postural stability and coordination [53]; gait speed and functional capacity [56].” – It is important to pay special attention to the points that refer greater benefits regarding gait and postural stability.

On the other hand, the authors intend to take into account not only the theoretical framework currently recognized by the scientific community, but also to highlight other experimental conceptual references, namely those related to biomechanical characteristics, which in turn are related to the particularities of the people involved in the laboratory phases. The aim of this paper is to provide a solid theoretical support, combined with a laboratory basis, but not a "fit for all".

Page 7. Line 312. Why three workouts per week if the literature already shows that twice a week are enough to promote gains related to strength, functional capacity, balance and other parameters?

RESPONSE: Thank you for your comment. The proposal of a minimum of three sessions per week contrasts with some accepted theoretical and practical bases already published. We agree that twice a week may be sufficient, as the minimum recommended in the literature. However, we suggest that increasing the frequency of activity is not only sufficient but necessary to achieve the proposed aims. In this case, and from personal experience, which is not systematically controlled, it can be deduced that, for example, missing one session out of two per week corresponds to a longer stimulus interval, which may slow down the desired effect. Thus, less time in each session, but more weekly frequency, can help to create a positive feedback in maintaining the gains made by the activity and a positive inertia in participation, providing the desired positive effects in everyday life. This logical factor introduced in the constructed of the intervention model is related to the general conceptual issue of the paper. Knowing and taking into account what has been published, but creating more logical-deductive foundations subject to controlled laboratory experimentation and supervised by an ethics committee.

Complementary, the authors mention in the article (line 115): “Nonetheless, the optimal dose–response of this type of exercises is yet to be defined due to the enormous variability and lack of detail observed in the various researches concerning the training parameters, e.g., weekly frequency and workout duration. Despite this fact, it is widely accepted that stimuli applied to nervous structures during exercise will be more effective in conditions of absence of fatigue [44]”. (line 180) “The National Academy of Sports Medicine (NASM) postulated that exercises progression with elderly should be slow, well monitored, and based on postural control [58].”  

Present a limitations session, indicating whether this training would be suitable for all elderly people, considering different age groups, pathologies, etc.

RESPONSE: Thank you for your comment. The following paragraph was added:

“3.1. Limitations of the exercise program proposal and special considerations

The implementation of the exercise program proposal may encompass some difficulties in very frail elderly people. Nonetheless, remembering that this proposal is just that and not a recipe, in very frail elderly the exercise prescription should be based on four movement pillars, especially in the locomotion pillar; even if this means reducing exercises complexity to a minimum – i.e., single-joint exercises with maximum stability. In this way, the optimal situation would be one-to-one sessions, e.g., sessions run by “personal trainers”. On the other hand, with three shorter sessions a week it is also possible to organize groups with fewer people and according to specific needs (for psycho-motor development, pathologies, etc.). The model must therefore be adapted to each of these groups, with less dispersion of attention on the part of the professional leading the session.”

Reviewer 2 Report

Comments and Suggestions for Authors

The authors address the problem of falling among the Ederly. The question he wants to provide some answers to is: “What types of exercises should be included in exercise programs to prevent falls? The aim of this manuscript was to present guidelines for an exercise program to prevent falls among the elderly, created with the aim of improving biomechanical parameters of gait linked to falls.

The subject is interesting and fits in perfectly with the journal's topics.  The proposed exercise program draws on international literature to develop a structural framework. However, there is no quantified justification of the benefits and gains of such a program. I feel it is essential to consider the following questions and comments in order to improve the article:

-        -   Why not test the proposed exercise program on a sample of elderly people?

-         -  What biomechanical parameters could be evaluated to demonstrate the interest and potential gains of the program?

-         -  The duration of the program and exercises could be the subject of a study to justify the choices made and quantify the gains of the program.

-         -  Finally, section 2 could be greatly reduced in favor of a results paragraph highlighting the gains achieved by the proposed exercise program.

Author Response

Dear reviewer.

Thank you very much for revisions and suggestions. We truly believe that this revisions and suggestions will improve the manuscript.

The authors address the problem of falling among the Ederly. The question he wants to provide some answers to is: “What types of exercises should be included in exercise programs to prevent falls? The aim of this manuscript was to present guidelines for an exercise program to prevent falls among the elderly, created with the aim of improving biomechanical parameters of gait linked to falls.

RESPONSE: Thank you for your comment. This is a very important observation. In fact, the title can introduce conceptual noise. The review aim is to provide a basis for 'guidelines' to be followed, not 'recipes' for exercises ('exercise programs'). That's why, throughout the text, the authors do not propose exercise prescriptions or specific biomechanical parameters for each group of performers. The "guidelines" or recommendations to be followed should therefore be adapted to each group. The same principle applies to the specificity and limits of variability of the biomechanical parameters to be studied. Thank you again for your comment, and to avoid this, we propose to change the title to: “Proprioceptive and strength exercise guidelines to prevent falls in the elderly and related to biomechanical movement characteristics”. On the other hand, it should be emphasized that the authors propose: a) line 22, end of abstract "Based on theoretical rationale, a proposal for a fall prevention exercise program was presented". b) line 60, end of introduction "... the explanation of its theoretical rationale and development was obviously included". Line 62, title of Chapter 2: "Theoretical Rationale of the Exercise Program". So the focus of this review is on the theoretical rationale and not on the prevention exercise program.

The subject is interesting and fits in perfectly with the journal's topics.  The proposed exercise program draws on international literature to develop a structural framework. However, there is no quantified justification of the benefits and gains of such a program. I feel it is essential to consider the following questions and comments in order to improve the article:

RESPONSE: Thank you for your comment. Following the previous answer, the authors point out in the abstract and text (line 56) that: “The critical review performed during this manuscript collected important evidence and knowledge in order to create a structural basis for the development of a fall prevention exercise program.”. Therefore, authors do not aim to present a recipe for an exercise program. The program will be based on the fundamentals that the review essentially presents in section "2."

Why not test the proposed exercise program on a sample of elderly people?

RESPONSE: Thank you for your comment. Here too, the previous answers justify this apparent absence. Authors just want to explain what they consider to be a "structural framework" and extend the guidelines based on section "2." to the scientific community that works with different types of older people. If the community works as a network, the "structural framework" will be more consistent, and the results of its applications (specific populations) can also provide more indications and evaluation criteria through biomechanical parameters (their limits and variability) of the people who practice.

What biomechanical parameters could be evaluated to demonstrate the interest and potential gains of the program?

RESPONSE: Thank you for your comment. Biomechanical parameters and their variability translate and correspond to the externalization of how the performer controls their movements and forces. Through this concept, laboratory evaluations take into account "normal" biomechanical parameters and their variability. Thus, the continued quantitative monitoring of the chosen biomechanical parameters (the paper presents the respective "guidelines", line 202) will advocate the "potential gains of the program". In this way, the following point was added:

“4. Conclusions and Future Investigation

(…)

Future investigation should consider testing the guidelines presented regarding a fall prevention exercise program in elderly. In this way, the referred biomechanical gait parameters associated with falls should be evaluated using gold standard measures, i.e., three-dimensional motion analysis systems [68].”

The duration of the program and exercises could be the subject of a study to justify the choices made and quantify the gains of the program.

RESPONSE: Thank you for your comment. Authors agree. Only with the construction of each experimental case (it follows the guidelines, has its own prescriptions and therefore specific laboratory evaluations) can the laboratory framework for data collection and its conclusion be defined. In this way, the following sentence was added to the section Conclusions and Future Investigation: “On the other hand, the duration of the program and exercises could be the subject of a study to justify the choices made and quantify the gains of the program.”

Finally, section 2 could be greatly reduced in favor of a results paragraph highlighting the gains achieved by the proposed exercise program.

RESPONSE: Thank you for your comment. According to the previous responses, the authors consider the development of section "2." is appropriate to the objectives of this review. The text represents a logical-deductive set of the model that led to the guidelines and which will be adapted to each exercise program prescription.

Round 2

Reviewer 2 Report

Comments and Suggestions for Authors

The authors responded to comments and questions. I accept the article in its current form.

Sincerely